# IMPROVED GENERALIZATION BOUND OF PERMUTATION INVARIANT DEEP NEURAL NETWORKS

## ABSTRACT

We theoretically prove that learning with permutation invariant deep neural networks largely improves their generalization performance. Learning problems with data that are invariant to permutations are frequently observed in various applications, for example, point cloud data and graph data. Numerous methodologies have been developed and they achieve great performances, however, understanding a mechanism of the performance is still a developing problem. In this paper, we derive a theoretical generalization bound for invariant deep neural networks with a ReLU activation to clarify their mechanism. Consequently, our bound shows that the main term of their generalization gap is improved by $\sqrt{n!}$ where $n$ is a number of permuting coordinates of data. Moreover, we prove that an approximation power of invariant deep neural networks can achieve an optimal rate, though the networks are restricted to be invariant. To achieve the results, we develop several new proof techniques such as correspondence with a fundamental domain and a scale-sensitive metric entropy.

## 1 INTRODUCTION

A learning task with permutation invariant data frequently appears in various situations in data analysis. A typical example is learning on sets such as a point cloud, namely, the data are given as a set of points and permuting the points in the data does not change a result of its prediction. Another example is learning with graphs which contain a huge number of edges and nodes. Such the tasks are very common in various scientific fields (Ntampaka et al., 2016; Ravanbakhsh et al., 2016; Faber et al., 2016), hence, numerous deep neural networks have been developed to handle such the data with invariance (Zaheer et al., 2017; Li et al., 2018a; Su et al., 2018; Li et al., 2018b; Yang et al., 2018; Xu et al., 2018). The succeeding methods show that their networks for invariance can greatly improve the accuracy with a limited size of networks and data.

An important question with invariant data is to understand the reason for the empirical high accuracy from theoretical aspects. Since invariant data are high-dimensional in general, learning theory claims that the high-dimensionality reduces its generalization performance. However, the methods for invariant data achieve better accuracy, thus it contradicts the theoretical principle. Though several theoretical studies (Maron et al. (2019) and Sannai et al. (2019)) prove a universal approximation property of neural networks for invariant data and guarantee that invariant deep neural networks have sufficient expressive power, the generalization power of the invariant deep neural networks is left as an open question.

In this paper, we prove a theoretical bound for generalization of invariant deep neural networks. To show an overview of our result, we provide a simplified version as follows. We consider a supervised-learning problem with $m$ pairs of observations $(X_i, Y_i)$ where $X_i$ are regarded as $p$-dimensional vectors, and $X_i$ can divided to $n$ coordinates and each of them have $D = p/n$ dimension. Also, let $f^{S_n}$ denote a function by a deep neural network which satisfies an invariant property, $f(x) = f(\sigma \cdot x)$ holds for any $x \in \mathbb{R}^{n \times D}$ where $\sigma$ is an arbitrary permutation of $D$-dimensional coordinates in $x$. Also, we define $R_m(f) = m^{-1} \sum_{i=1}^{m} L(Y_i, f(X_i))$ and $R(f) = \mathbb{E}[L(Y, f(X))]$ as an empirical and expected loss value $L(Y, f(X))$. Then, we show that following:

**Theorem 1** (Informal version of Theorem 2). *Let $f^{S_n}$ be a function by a deep neural network which takes $p$-dimensional inputs and invariant to any permutations of $n$ coordinates. Then, for sufficiently*

*small $\varepsilon > 0$, we obtain*

$$R(f^{S_n}) \le R_m(f^{S_n}) + \sqrt{\frac{C}{n!\, m\varepsilon^p}} + O(\log(1/\varepsilon)),$$

*with probability at least $1 - O(\varepsilon)$. Here, $C > 0$ is a constant independent of $m$ and $n$.*

As a consequence of Theorem 1, the generalization bound is improved by $\sqrt{n!}$ by the invariant property. Since the number of coordinates $n$ is huge in practice, e.g. there are $n \ge 1,000$ points in the point cloud data in Zaheer et al. (2017) and hence $\sqrt{n!} \ge 10^{1,000}$ holds, we show that the derived generalization bound is largely improved by invariance. Further, we also derive a rate of approximation of neural networks with invariance (Theorem 4) and its optimality, thus we show that an invariance property for deep neural networks does not reduce an expressive power.

From a technical aspect, we develop mainly three proof technique to obtain the improved bound in Theorem 1. Firstly, we introduce a notion of a *fundamental domain* to handle invariance of functions and evaluate the complexity of the domain (Lemma 1). Secondly, we show a one-to-one correspondence between a function by invariant deep neural networks and a function on the fundamental domain (Proposition 2). Thirdly, we develop a *scale-sensitive covering number* to control a volume of invariant functions with neural networks (Proposition 5). Based on the techniques, we can connect a generalization analysis to the invariance of deep neural networks.

We summarize the contributions of this paper as follow:

- We investigate the generalization bound of deep neural networks which are invariant to permutation of $n$ coordinates, then we show that the bound is improved by $\sqrt{n!}$.

- We derive a rate of approximation of invariant deep neural networks. The result shows that the approximation rate is optimal.

- We develop several proof techniques to achieve the bound such as a complexity analysis for a fundamental domain and a scale-sensitive metric entropy.

## 1.1 NOTATION

For a vector $b \in \mathbb{R}^D$, its $d$-th element is denoted by $b_d$. Also, $b_{-d} := (b_1, ..., b_{d-1}, b_{d+1}, ..., b_D) \in \mathbb{R}^{D-1}$ is a vector without $b_d$. $\|b\|_q := (\sum_{j=d}^D b_d^q)^{1/q}$ is the $q$-norm for $q \in [0, \infty]$. For a tensor $A \in \mathbb{R}^{D_1 \times D_2}$, a $(d_1, d_2)$-th element of $A$ is written as $A_{d_1, d_2}$. For a function $f : \Omega \to \mathbb{R}$ with a set $\Omega$, $\|f\|_{L^q} := (\int_\Omega |f(x)|^q dx)^{1/q}$ denotes the $L^q$-norm for $q \in [0, \infty]$. For a subset $\Lambda \subset \Omega$, $f_{\restriction \Lambda}$ denotes the restriction of $f$ to $\Lambda$. For an integer $z$, $z! = \prod_{j=1}^n j$ denotes a factorial of $z$. For a set $\Omega$ with a norm $\| \cdot \|$, $\mathcal{N}(\varepsilon, \Omega, \| \cdot \|) := \inf\{N : \exists\{\omega_j\}_{j=1}^N \text{ s.t. } \cup_{j=1}^N \{\omega : \|\omega - \omega_j\| \le \varepsilon\} \supset \Omega\}$ is a covering number of $\Omega$ with $\varepsilon > 0$. For a set $\Omega$, $\mathrm{id}_\Omega$ or $\mathrm{id}$ denotes the identity map on $\Omega$, namely $\mathrm{id}_\Omega(x) = x$ for any $x \in \Omega$. For a subset $\Delta \subset \mathbb{R}^n$, $\mathrm{int}(\Delta)$ denotes the set of the inner points of $\Delta$.

## 2 PROBLEM SETTING

### 2.1 INVARIANT DEEP NEURAL NETWORK

We define a set of permutation $S_n$ in this paper. Consider $x \in \mathbb{R}^{n \times D}$ where $n$ be a number of coordinates in $x$ and $D$ be a dimension of each coordinate. Then, an action $\sigma \in S_n$ on $x$ is defined as

$$(\sigma \cdot x)_{i,d} = x_{\sigma^{-1}(i),d}, \ i = 1, ..., n, d = 1, ..., D,$$

here, $\sigma$ is a permutation of indexes $i$. Also, we define an invariant property for general functions.

**Definition 1** ($S_n$-Invariant/Equivariant Function). For a set $\mathcal{X} \subset \mathbb{R}^{n \times D}$, we say that a map $f : \mathcal{X} \to \mathbb{R}^M$ is

- $S_n$-*invariant* (or simply *invariant*) if $f(\sigma \cdot x) = f(x)$ for any $\sigma \in S_n$ and any $x \in \mathcal{X}$,

- $S_n$-*equivariant* (or simply *equivariant*) if there is an $S_n$-action on $\mathbb{R}^M$ and $f(\sigma \cdot x) = \sigma \cdot f(x)$ for any $\sigma \in S_n$ and any $x \in \mathcal{X}$.

In this paper, we mainly treat fully connected deep neural networks with a ReLU activation function. The ReLU activation function is defined by $\mathrm{ReLU}(x) = \max(0, x)$. Deep neural networks are built by stacking blocks which consist of a linear map and a ReLU activation. More formally, it is a function $Z_i : \mathbb{R}^{d_i} \to \mathbb{R}^{d_{i+1}}$ defined by $Z_i(x) = \mathrm{ReLU}(W_i x + b_i)$, where $W_i \in \mathbb{R}^{d_{i+1} \times d_i}$ and $b_i \in \mathbb{R}^{d_{i+1}}$ for $i = 1, ..., H$. Here, $H$ is a depth of the deep neural network and $d_i$ is a width of the $i$-th layer. An output of deep neural networks is formulated as

$$f(x) := Z_H \circ Z_{H-1} \ldots Z_2 \circ Z_1(x). \tag{1}$$

Let $\mathcal{F}_{DNN}$ be a set of functions by deep neural networks.

We also consider an invariant deep neural network defined as follows:

**Definition 2** (Invariant Deep Neural Network). $f \in \mathcal{F}_{DNN}$ is a function by a $S_n$-*invariant deep neural network*, if $f$ is a $S_n$-invariant function. Let $\mathcal{F}_{DNN}^{S_n} \subset \mathcal{F}_{DNN}$ be a set of functions by $S_n$-invariant deep neural networks.

The definition is a general notion and it contains several explicit invariant deep neural networks. We provide several representative examples as follow.

**Example 1** (Deep Sets). Zaheer et al. (2017) develops an architecture for invariant deep neural networks by utilizing layer-wise equivariance. Their architecture consists of equivariant layers $\ell_1, ..., \ell_j$, an invariant linear layer $h$, and a fully-connected layer $f'$. For each $\ell_i.i = 1, .., j$, its parameter matrix is defined as

$$W_i = \lambda \boldsymbol{I} + \gamma(\boldsymbol{1}\boldsymbol{1}^\top), \lambda, \gamma \in \mathbb{R}, \boldsymbol{1} = [1, ..., 1]^\top,$$

which makes $\ell_i$ as a layer-wise equivariant function. They show that $f = f' \circ h \circ \ell_j \circ \cdots \ell_1$ is an invariant function. Its illustration is provided in Figure 1.

**Example 2** (Invariant Feature Extraction). Let $e$ is a mapping for invariant feature extraction which will be explicitly constructed by deep neural networks in Proposition 2. Then, a function $f = g \circ e$ where $g$ is a function by deep neural networks with a restricted domain. Figure 2 provides its image.

## 2.2 Learning Problem with Invariant Network

**Problem formulation**: Let $I = [0, 1]^{n \times D}$ be an input space with dimension $p = dD$. Let $\mathcal{Y}$ be an output space. Also, let $L : \mathcal{Y} \times \mathcal{Y} \to \mathbb{R}$ be a loss function which satisfies $\sup_{y,y' \in \mathcal{Y}} |L(y, y')| \leq 1$ and 1-Lipschitz continuous. Let $P^*(x, y)$ be the true unknown distribution on $I \times \mathcal{Y}$, and for $f : I \to \mathcal{Y}$, $R(f) = \mathbb{E}_{(X,Y) \sim P^*}[L(f^*(X), Y)]$ be the expected risk of $f$. Also, suppose we observe a training dataset $\mathcal{D}_m := \{(X_1, Y_1), ..., (X_m, Y_m)\}$ of size $m$. Let $R_m(f) := m^{-1} \sum_{i=1}^m L(f(X_i), Y_i)$ be the empirical risk of $f$. A goal of this study to investigate the expected loss $R(f)$ with a function $f$ from a set of functions as a hypothesis set.

**Learning with Invariant Network**: We consider learning with a hypothesis set by invariant deep networks. Namely, we fix an architecture of deep neural networks preserves $f^{S_n} \in \mathcal{F}_{DNN}^{S_n}$ to be an invariant function. Then, we evaluate the expected loss $R(f^{S_n})$.

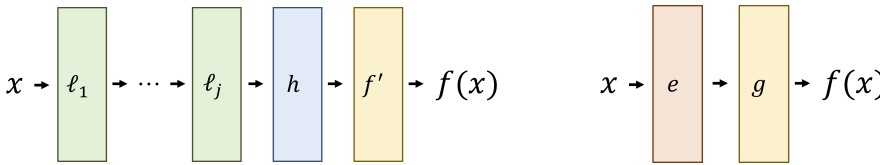

Figure 1: The invariant deep neural network by DeepSets (Zaheer et al. (2017)). $\ell$ is an equivariant layer, $h$ is a linear invariant layer, and $f'$ is a function by networks.

Figure 2: The invariant deep neural network by a fully connected layer $g$ and a feature extraction layer $e$.

# 3 MAIN RESULT

## 3.1 COMPLEXITY-CONTROL BOUND

We show that the learning procedure with invariance can largely improve the generalization performance of a deep neural network by proving the improved bound for the generalization error of $\hat{f}$ with invariance. We firstly derive a *Complexity-dependent bound* which holds with an arbitrary true distribution. The bound depends on a Complexity control of $\mathcal{F}_{DNN}^{S_n}$ and the Rademacher complexity.

**Theorem 2** (Main Result 1). *Let $\mathcal{F}_{DNN}^{S_n}$ be a set of functions by $S_n$-invariant deep neural networks which are $C_\Delta$-Lipschitz continuous and bounded by $B > 0$. Then, for any $f^{S_n} \in \mathcal{F}_{DNN}^{S_n}$ and for any $\varepsilon > 0$, the following inequality holds with probability at least $1 - 2C_\Delta\varepsilon$:*

$$R(f^{S_n}) \leq R_m(f^{S_n}) + \underbrace{\sqrt{\frac{2c_1}{n!\, m\varepsilon^p}}}_{=:I_1} + \underbrace{\sqrt{\frac{2\log(2c_2 B/\varepsilon) + 2\log(1/2\varepsilon)}{m}}}_{=:I_2},$$

*where $c_1, c_2 > 0$ are constants which are independent of $n$ and $m$.*

Significantly, the main term $I_1$ in Theorem 2 is improved by $\sqrt{n!}$ in the denominator. Note that we regard $I_1$ as the main term since $I_2$ is a logarithmic order in $\varepsilon$. As $n$ is huge in practice, e.g. a number of points in point cloud data, the term $n!$ largely improves the tightness of the bound.

We mention that $\varepsilon^p$ appears in the denominator of $I_1$ and it looks a harmful term when $p = nD$ is large. However, the term is negligible with large $n$, because $n!$ increases much faster than $\varepsilon^{nD}$. Namely, we can obtain $n!\varepsilon^{nD} = \Omega(n^c)$ holds for any $c > 0$ and $\varepsilon > 0$ as $n \to \infty$.

Proof of Theorem 2 utilizes a complexity control for $\mathcal{F}_{DNN}^{S_n}$. As a preparation, we apply the well-known bound (e.g. a slightly modified version of Theorem 10.1 in Anthony & Bartlett (2009)) and obtain

$$R(f^{S_n}) \leq R_m(f^{S_n}) + \sqrt{\frac{2\log 2\mathcal{N}(\varepsilon, \mathcal{F}_{DNN}^{S_n}, \|\cdot\|_{L^\infty}) + 2\log(1/2\varepsilon)}{m}}, \tag{2}$$

which describes generalization of $f^{S_n}$ by the covering number $\log 2\mathcal{N}(\varepsilon, \mathcal{F}_{DNN}^{S_n}, \|\cdot\|_{L^\infty})$. Then, we bound the covering number by the following Theorem which plays a key role to achieve the main result in Theorem 2. Proof of Theorem 3 depends on several newly developed results presented in Section 4.

**Theorem 3** (Complexity Bound). *Let $\mathcal{F}_{DNN}^{S_n}$ be defined in section 2. Then, with an existing constant $c > 0$, we obtain*

$$\log \mathcal{N}(2C_\Delta\delta, \mathcal{F}_{DNN}^{S_n}, \|\cdot\|_{L^\infty(I)}) \leq \frac{c}{n!\,\delta^p} + \log\left(\frac{2cB}{\delta}\right).$$

**Remark 1** (Bound without invariance). The bound is a general version of an ordinary learning $f \in \mathcal{F}_{DNN}$ which does not have invariance. Rigorously, suppose $\mathcal{F}_{DNN}$ is a set of functions which are $C_\Delta$-Lipschitz continuous and bounded by $B > 0$. Then, for any $f \in \mathcal{F}_{DNN}$ and $\varepsilon > 0$, the inequality in Theorem 2 holds with $n = 1$.

**Remark 2** (Bound for covering numbers). We mention that there is another way to bound the covering number of $\mathcal{F}_{DNN}^{S_n}$ by a number of parameters (e.g. Theorem 14.5 in Anthony & Bartlett (2009)). Such a bound has a fast order since its order is a logarithm of $\varepsilon$. However, the bound has a linear order in a number of parameters, hence it easily increases with large-scale deep neural networks which possess a huge number of nodes and edges. Moreover, such a bound is independent of the volume of the domain, hence we cannot obtain the scale-sensitive covering number. To avoid the problem, we employ another strategy in Theorem 3.

## 3.2 APPROXIMATION-CONTROL BOUND

We investigate the approximation power of invariant deep neural networks to clarify how they can achieve a small empirical loss. Although we restrict the expressive power of deep neural networks

in the learning procedure, we prove that our networks have sufficient power of approximation. To the aim, we define the *Hölder space* which is a class of smooth functions, then investigate the approximation power of invariant deep neural networks for the space.

**Definition 3** (Hölder space). Let $\alpha > 0$ be a degree of smoothness. For $f : I \to \mathbb{R}$, the *Hölder norm* is defined as

$$\|f\|_{\mathcal{H}^\alpha} := \max_{\beta:|\beta|<\lfloor\alpha\rfloor} \sup_{x\in I} |\partial^\beta f(x)| + \max_{\beta=\lfloor\alpha\rfloor} \sup_{x,x'\in I, x\neq x'} \frac{|\partial^\beta f(x) - \partial^\beta f(x')|}{\|x-x'\|_\infty^{\alpha-\lfloor\alpha\rfloor}}.$$

Then, the *Hölder space* on $I$ is defined as

$$\mathcal{H}^\alpha = \left\{ f \in C^{\lfloor\alpha\rfloor} \,\Big|\, \|f\|_{\mathcal{H}^\alpha} < \infty \right\}.$$

Also, $\mathcal{H}_B^\alpha = \{f \in \mathcal{H}^\alpha \mid \|f\|_{\mathcal{H}^\alpha} \leq B\}$ denotes the $B$-radius closed ball in $\mathcal{H}^\alpha$.

Intuitively, $\mathcal{H}^\alpha$ is a set of bounded functions which are $\alpha$-times differentiable. The notion of the Hölder space is often utilized in characterizing the optimal functions $f^*$ (e.g. see Schmidt-Hieber (2017)). We achieve the more detailed bound for the generalization error with assuming $f^* \in \mathcal{H}_B^\alpha$.

**Theorem 4** (Main Theorem 2). *For any $\varepsilon > 0$, suppose $\mathcal{F}_{DNN}^{S_n}$ has at most $\mathcal{O}(\log(1/\varepsilon))$ layers and $\mathcal{O}(\varepsilon^{-p/\alpha}\log(1/\varepsilon))$ non-zero parameters. Then, for any invariant $f^* \in \mathcal{H}_B^\alpha$, there is $f^{S_n} \in \mathcal{F}_{DNN}^{S_n}$ such that*

$$\|f^{S_n} - f^*\|_{L^\infty(I)} \leq \varepsilon.$$

The result in Theorem 4 clarifies the approximation power of deep networks, and also show that a sufficient number of parameters (nodes) makes the generalization error converge to zero. Also, the theorem shows that the approximation error decreases as the number of parameters increase with the rate $-p/\alpha$ up to log factors. The rate is the optimal rate by Yarotsky (2017) without invariance. Hence, we prove that the deep networks with invariance can achieve the optimal approximation rate even with the invariance restriction.

## 4 PROOF AND ITS STRATEGY

### 4.1 FUNDAMENTAL DOMAIN AND ITS CORRESPONDENCE

To handle the invariance property in our proof, we provide a key notion to show the main result.

**Definition 4** (Fundamental Domain). Let $G$ be a group acting on a set $J$. $\Delta \subset J$ is said to be a *fundamental domain* of $J$ with respect to the action of $G$ if $\Delta$ satisfy the following properties;

- $J = \cup_{\sigma \in G} \{\sigma \cdot x \mid x \in \Delta\}$.
- $\sigma \cdot \text{int}(\Delta) \cap \tau \cdot \text{int}(\Delta) = \emptyset$ for any $\sigma \neq \tau \in G$.

In our case, we can take a fundamental domain explicitly.

**Proposition 1.** *Put $I = [0,1]^{n\times D}$. Then*

$$\Delta := \{x \in I \mid x_{1,1} \geq x_{2,1} \geq \cdots \geq x_{n,1}\}$$

*is a fundamental domain of $I$ with respect to the permutation action of $S_n$ defined in Section 2.1.*

Figure 3 provides $\Delta$ with $n = 3$ and $D = 1$. Intuitively, $\Delta$ is an extracted feature space for an invariant function. Any element of $I$ corresponds to an element of $\Delta$ with an existing action in $S_n$, namely, we can obtain

$$I = \cup_{\sigma \in S_n} \{\sigma \cdot x \mid x \in \Delta\}.$$

*Proof of Proposition 1.* We confirm the first property of the fundamental domain, namely $I = \cup_{\sigma \in G} \{\sigma \cdot x \mid x \in \Delta\}$. Take $x \in I$. There is a $\sigma^{-1} \in S_n$ such that $x_{\sigma(1),1} \geq x_{\sigma(2),1} \geq \cdots \geq x_{\sigma(n),1}$. Then by the definition of $\Delta$, $\sigma^{-1} \cdot x \in \Delta$. Hence $x \in \sigma \cdot \Delta = \{\sigma \cdot x \mid x \in \Delta\}$. This

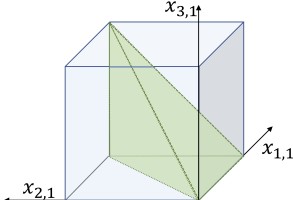 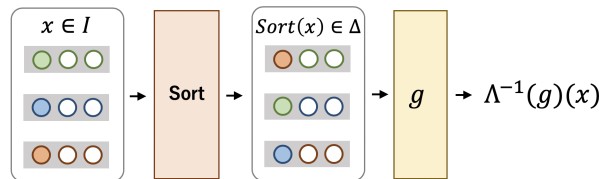

Figure 3: A fundamental domain $\Delta$ (the green cone) in $I$ (the blue cube) with $n = 3$ and $D = 1$.

Figure 4: The Sort layer (red) which converts $g \in \mathcal{F}^{\Delta}_{DNN}$ to $f = \Lambda^{-1}(g) \in \mathcal{F}^{S_n}_{DNN}$ with $n = D = 3$. The Sort layer exchanges the first elements of each $x_i \in \mathbb{R}^D$.

implies the first property.

We confirm the second property. We have $\text{int}(\Delta) = \{x \in I \mid x_{1,1} > x_{2,1} > \cdots > x_{n,1}\}$ By the definition of our action, $\sigma \cdot \text{int}(\Delta) = \{x \in I \mid x_{\sigma^{-1}(1),1} > x_{\sigma^{-1}(2),1} > \cdots > x_{\sigma^{-1}(n),1}\}$. Hence $\sigma \cdot \text{int}(\Delta) \cap \tau \cdot \text{int}(\Delta) = \emptyset$ for any $\sigma \neq \tau \in G$. $\qquad \square$

We provide two important properties of $\Delta$. Firstly, we start with showing that there is a one-to-one correspondence between deep neural networks on $\Delta$ and invariant deep neural networks on $I$. To the aim, we consider a set of functions on $\Delta$ by (not necessarily invariant) deep neural networks;

$$\mathcal{F}^{\Delta}_{DNN} = \{g : \Delta \to \mathbb{R} \mid g \text{ has the form (1)}\}.$$

Then, we obtain the following result:

**Proposition 2.** *There exists a bijection map $\Lambda : \mathcal{F}^{S_n}_{DNN} \to \mathcal{F}^{\Delta}_{DNN}$. Further, for any $f \in \mathcal{F}^{S_n}_{DNN}$, $\Lambda(f)$ is obtained by the restriction of $f$, namely $\Lambda(f) = f_{\upharpoonright \Delta}$, and for any $g \in \mathcal{F}^{\Delta}_{DNN}$, $\Lambda^{-1}(g)$ can be obtained by adding **sorting** layers appeared in the proof.*

Figure 4 provides an image for $\Lambda^{-1}(g)$ for $g \in \mathcal{F}^{\Delta}_{DNN}$. For preparation for proof of Proposition 2, we define an explicit invariant deep neural network. For a vector $z \in \mathbb{R}^N$ for some $N$, let $\max^{(j)}(z_1, \ldots, z_N)$ (resp. $\min^{(j)}(z_1, \ldots, z_N)$) be a function which returns the $j$-th largest (resp. smallest) element of $\{z_1, \ldots, z_N\}$. We can easily see that these functions are a $S_n$-invariant function. More strongly, we have the following proposition.

**Proposition 3.** $\max^{(j)}(z_1, \ldots, z_N)$ *and* $\min^{(j)}(z_1, \ldots, z_N)$ *are represented by an existing deep neural networks with an ReLU activation for any $j = 1, ..., N$.*

*Proof of Proposition 3.* Firstly, since

$$\max(z_1, z_2) = \max(z_1 - z_2, 0) + z_2, \text{ and } \min(z_1, z_2) = -\max(z_1 - z_2, 0) + z_1$$

hold, we see the case of $j = 1, N = 2$. By repeating $\max(z_1, z_2)$, we construct $\max^{(1)}(z_1, \ldots, z_N)$ and $\min^{(1)}(z_1, \ldots, z_N)$. Namely, we prove the claim in the case of $j = 1$ and arbitrary $N$. At first, we assume $N$ is even without loss of generality, then we divide the set $\{z_1, ... z_N\}$ into sets of pairs $\{(z_1, z_2), ... (z_{N-1}, z_N)\}$. Then, by taking a max operation for each of the pairs, we have $\{y_1 = \max(z_1, z_2), ..., y_{N/2} = \max(z_{N-1}, z_N)\}$. We repeat this process to terminate. Then we have $\max^{(1)}(z_1, \ldots, z_N)$ it is represented by an existing deep neural network. Similarly, we have $\min^{(1)}(z_1, \ldots, z_N)$. Finally, we prove the claim on $j = 2, ..., N$ by induction. Assume that for any $N$ and $\ell < j$, $\max^{(\ell)}(z_1, \ldots, z_N)$ is represented by a deep neural network. We construct $\max^{(j)}(z_1, \ldots, z_N)$ as follows: since

$$\max^{(j-1)}(z_{-\ell}) = \begin{cases} \max^{(j-1)}(z_1, \ldots, z_N) & (\text{if } z_\ell \leq \max^{(j)}(z_1, \ldots, z_N)) \\ \max^{(j)}(z_1, \ldots, z_N) & (\text{otherwise}) \end{cases}$$

holds, we have $\max^{(j)}(z_1, \ldots, z_N) = \min(\{\max^{(j-1)}(Z_\ell) \mid \ell = 1, ..., N\})$. By inductive hypothesis, the right hand side is represented by a deep neural network. $\qquad \square$

*Proof of Proposition 2.* We first define sorting layers which is an $S_n$-invariant network mapping from $I$ to $\Delta$. When $D = 1$, put $\text{Sort}_1(x_{1,1}, \ldots, x_{n,1}) = (\max^{(1)}(x_{1,1}, \ldots, x_{n,1}), \ldots, \max^{(n)}(x_{1,1}, \ldots, x_{n,1}))$. Then by Proposition 3, $\text{Sort}_1(x_{1,1}, \ldots, x_{n,1}) = \text{Sort}_1(x_{1,1}, \ldots, x_{n,1})$ is also a function by an $S_n$-invariant deep neural network and $\text{Sort}(x_{1,1}, \ldots, x_{n,1})$ is the function from $I$ to $\Delta$. When $D > 1$, we first consider $\text{Sort}_1(x_{1,1}, \ldots, x_{n,1})$. Since $\text{Sort}_1(x_{1,1}, \ldots, x_{n,1})$ gives a permutation on $(x_{1,1}, \ldots, x_{n,1})$, for each $(x_{1,1}, \ldots, x_{n,1})$, we can find $\sigma \in S_n$ such that

$$\text{Sort}_1(x_{1,1}, \ldots, x_{n,1}) = (\text{Sort}_1(x_{1,1}, \ldots, x_{n,1})_1, \ldots, \text{Sort}_1(x_{1,1}, \ldots, x_{n,1})_n) = (x_{\sigma(1),1}, \ldots, x_{\sigma(n),1}).$$

Then we define

$$\text{Sort}(x) = \begin{pmatrix} \text{Sort}_1(x_{1,1}, \ldots, x_{n,1})_1 & \cdots & x_{\sigma(1),d} & \cdots & x_{\sigma(1),D} \\ \vdots & \ddots & & & \vdots \\ \text{Sort}_1(x_{1,1}, \ldots, x_{n,1})_i & & x_{\sigma(i),d} & & x_{\sigma(i),D} \\ \vdots & & & \ddots & \vdots \\ \text{Sort}_1(x_{1,1}, \ldots, x_{n,1})_n & \cdots & x_{\sigma(n),d} & \cdots & x_{\sigma(n),D} \end{pmatrix}.$$

By the construction and the definition of $\Delta$, $\text{Sort}(x)$ is the function to $\Delta$. We confirm $\text{Sort}(x)$ is $S_n$-invariant. Take arbitrary $\tau \in S_n$ and fix $x$ and $\sigma \in S_n$ as above. Put $\tau \cdot x = y$. We show $\text{Sort}(y) = \text{Sort}(x)$. Since $\text{Sort}_1$ is an $S_n$-invariant function, we see $\text{Sort}_1(y_{1,1}, \ldots, y_{n,1}) = \text{Sort}_1(\tau(x_{1,1}, \ldots, x_{n,1})) = (x_{\sigma(1),1}, \ldots, x_{\sigma(n),1}) = (y_{\sigma(\tau^{-1}(1)),1}, \ldots, y_{\sigma(\tau^{-1}(n)),1})$. Then we have

$$\text{Sort}(y) = \begin{pmatrix} \text{Sort}_1(y_{1,1}, \ldots, y_{n,1})_1 & \cdots & y_{\sigma(\tau^{-1}(1)),d} & \cdots & y_{\sigma(\tau^{-1}(1)),D} \\ \vdots & \ddots & & & \vdots \\ \text{Sort}_1(y_{1,1}, \ldots, y_{n,1})_i & & y_{\sigma(\tau^{-1}(i)),d} & & y_{\sigma(\tau^{-1}(i)),D} \\ \vdots & & & \ddots & \vdots \\ \text{Sort}_1(y_{1,1}, \ldots, y_{n,1})_n & \cdots & y_{\sigma(\tau^{-1}(n)),d} & \cdots & y_{\sigma(\tau^{-1}(n)),D} \end{pmatrix}$$

$$= \begin{pmatrix} \text{Sort}_1(x_{1,1}, \ldots, x_{n,1})_1 & \cdots & x_{\sigma(1),d} & \cdots & x_{\sigma(1),D} \\ \vdots & \ddots & & & \vdots \\ \text{Sort}_1(x_{1,1}, \ldots, x_{n,1})_i & & x_{\sigma(i),d} & & x_{\sigma(i),D} \\ \vdots & & & \ddots & \vdots \\ \text{Sort}_1(x_{1,1}, \ldots, x_{n,1})_n & \cdots & x_{\sigma(n),d} & \cdots & x_{\sigma(n),D} \end{pmatrix}$$

$$= \text{Sort}(x),$$

where the second equality follows from $\tau^{-1} \cdot y = x$.

By using this function, we define the inverse of $\Lambda$. For any function $f$ by a deep neural network on $\Delta$, we define $\Phi(f) = f \circ \text{Sort}$. We confirm $\Lambda \circ \Phi = \text{id}_{\mathcal{F}_\Delta}$ and $\Phi \circ \Lambda = \text{id}_{\mathcal{F}^{S_n}}$. Since we have

$$\Lambda \circ \Phi(f) = \Lambda \circ f \circ \text{Sort} = (f \circ \text{Sort})_{\restriction \Delta} = f,$$

$\Lambda \circ \Phi$ is equal to $\text{id}_{\mathcal{F}_\Delta}$. Similarly,

$$\Phi \circ \Lambda(f) = \Phi \circ f_{\restriction \Delta} = f_{\restriction \Delta} \circ \text{Sort} = f,$$

where the last equality follows from the $S_n$-invariance of $f$. Hence, we have the desired result. $\square$

The second key property of $\Delta$ is that we can measure its size. Since $\Delta$ is included in $I$, we can naturally measure its volume by the Euclidean metric. By utilizing the property, we evaluate its volume by a covering number of $\Delta$ by the following lemma:

**Lemma 1** (Covering bound for $\Delta$). *There is a constant $C$ such that for enough small $\varepsilon > 0$, we obtain*

$$\mathcal{N}(\varepsilon, \Delta, \|\cdot\|_\infty) \leq \frac{C}{n! \, \varepsilon^{nD}}.$$

*Proof of Lemma 1.* Let $\mathcal{C}(I)$ be a set of $\varepsilon$-cubes which is a standard subdivision of $I$. We can easily see that $\mathcal{C}(I)$ attains the minimum value $\varepsilon^{-nD}$ of the number of $\varepsilon$-cubes which is the covering of $I$.

We show that we can find a subset of $\mathcal{C}(I)$ whose cardinality is $\frac{\varepsilon^{-nD}}{n!} + \mathcal{O}(\varepsilon^{-n(D-1)})$. The strategy of the proof is as follows. At first, we calculate the number $A$ of cubes in $\mathcal{C}(I)$ which intersect with the boundary of $\sigma \cdot \Delta$. Then since the permutation on the cubes which do not intersect with the boundary of $\sigma \cdot \Delta$ is free, if $A$ is $\mathcal{O}(\varepsilon^{-n(D-1)})$, we can find the covering whose cardinality is $\frac{\varepsilon^{-nD}}{n!} + \mathcal{O}(\varepsilon^{-n(D-1)})$. Since $\sigma \cdot \Delta$ is $\{x \in I \mid x_{\sigma^{-1}(1),1} \geq x_{\sigma^{-1}(2),1} \geq \cdots \geq x_{\sigma^{-1}(n),1}\}$, any boundary of $\sigma \cdot \Delta$ is of the form $\{x \in I \mid x_{\sigma^{-1}(1),1} \geq \cdots x_{\sigma^{-1}(i),1} = x_{\sigma^{-1}(i+1),1} \geq \cdots \geq x_{\sigma^{-1}(n),1}\}$.

Fix $\sigma$ and $i$. Consider the projection $\pi : \mathbb{R}^{n \times D} \to \mathbb{R}^{n-1 \times D}$ which sends $x_{\sigma^{-1}(i),1}$ to zero. $\pi$ induces the map $\widetilde{\pi} : \mathcal{C}(I) \to \mathcal{C}(\pi(I))$, where $\mathcal{C}(\pi(I))$ is a set of $\epsilon$-cubes which is the subdivision of $\pi(I)$ induced by $\mathcal{C}(I)$. Let $\mathcal{C}(I)_{diag}$ denote the set of cubes in $\mathcal{C}(I)$ which intersect with the set $B = \{x \in I \mid x_{\sigma^{-1}(i),1} = x_{\sigma^{-1}(i+1),1}\}$. Then we can see that $\widetilde{\pi}$ is injective on $\mathcal{C}(I)_{diag}$ as follows. Let us denote $\boldsymbol{a} = (a_{s,r}) \in \mathbb{R}^{nD}$ the center of an $\varepsilon$-cube in $\mathcal{C}(I)$. Assume that there are two cubes in $\mathcal{C}(I)_{diag}$ whose images by $\widetilde{\pi}$ are equal. Let us denote the centers of two cubes by $\boldsymbol{a}$ and $\boldsymbol{a}'$. Then we have $\widetilde{\pi}(\boldsymbol{a}) = \widetilde{\pi}(\boldsymbol{a}')$ and hence $a_{s,r} = a'_{s,r}$ holds for $(s,r) \neq (\sigma^{-1}(i),1)$. Here, by our construction of $\varepsilon$-cubes, a cube (in $\mathcal{C}(I)$) intersect with $B$ if and only if its center is on $B$. Therefore, since two cubes are in $\mathcal{C}(I)_{diag}$, we have $a_{\sigma^{-1}(i),1} = a_{\sigma^{-1}(i+1),1}$ and $a'_{\sigma^{-1}(i),1} = a'_{\sigma^{-1}(i+1),1}$. Hence $a_{s,r} = a'_{s,r}$ holds for any $(s,r)$ and two cubes are equal and $\widetilde{\pi}$ is injective on $\mathcal{C}(I)$.

Next, let $\mathcal{C}'(I)$ be the set of $\varepsilon$-cubes in $\mathcal{C}(I)$ which intersect a boundary of $\sigma \cdot \Delta$. We see that the cardinality of $\mathcal{C}'(I)$ is bounded by $E\varepsilon^{-n(D-1)}$ for some $E$. Since the number of components of the boundaries is finite, we prove the claim for a component of the boundary. Since $\widetilde{p}_{\restriction \mathcal{C}(I)_{diag}}$ is injective, we see the number of cubes which intersect the component is bounded by the number of $\varepsilon$-cubes in $\mathcal{C}(p(I))$, hence $\varepsilon^{-n(D-1)}$. Put $\mathcal{C}(I)_{inn} = \mathcal{C}(I) - \mathcal{C}'(I)$. Then each cubes in $\mathcal{C}(I)_{inn}$ does not intersect the boundaries of $\sigma \cdot \Delta$. Hence, there is a $\sigma$ such that the number of cubes $\mathcal{C}(I)_{inn}$ which are contained in $\sigma \cdot \Delta$ is lower than $\frac{|\mathcal{C}(I)_{inn}|}{n!}$. By adding the cubes which cover the boundaries of $\sigma \cdot \Delta$, we have the covering of $\sigma \cdot \Delta$. Furthermore, by pulling back by $\sigma$, we have the covering of $\Delta$. Hence, we have

$$\mathcal{N}(\varepsilon, \Delta, \|\cdot\|_\infty) \leq \frac{|\mathcal{C}(I)| - E\varepsilon^{-n(D-1)}}{n!} + E'\varepsilon^{-n(D-1)}.$$

Since $|\mathcal{C}(I)| = \varepsilon^{-nD}$, we have the desired result. $\qquad\square$

### 4.2 Proof for the Complexity-Control Bound (Theorem 2)

We utilize the results of $\Delta$ and prove Theorem 2. The proof mainly contains the following two-step: i) show that the covering number of $\mathcal{F}_{DNN}^{S_n}$ is equal to that of $\mathcal{F}_{DNN}^{\Delta}$, and ii) bound the covering number of $\mathcal{F}_{DNN}^{\Delta}$. The first step is provided by the following proposition.

**Proposition 4.** *For any $\varepsilon > 0$, we obtain*

$$\log \mathcal{N}(\varepsilon, \mathcal{F}_{DNN}^{S_n}, \|\cdot\|_{L^\infty(I)}) = \log \mathcal{N}(\varepsilon, \mathcal{F}_{DNN}^{\Delta}, \|\cdot\|_{L^\infty(I)}).$$

The result shows that the functional set by deep neural networks on $I$ with invariance is well described by a set of functions on $\Delta$ *without* invariance. The key point of this result is that we can describe the effect of invariance restriction on $\mathcal{F}_{DNN}^{S_n}$ by the size of $\mathcal{F}_{DNN}^{\Delta}$.

*Proof of Proposition 4.* For any $f, f' \in \mathcal{F}_{DNN}^{S_n}$, there exists $f_{\restriction \Delta}, f'_{\restriction \Delta} \in \mathcal{F}_\Delta$ by Proposition 2. Then, we can obtain

$$\|f - f'\|_{L^\infty(I)} = \|f_{\restriction \Delta} \circ g - f'_{\restriction \Delta} \circ g\|_{L^\infty(I)} \leq \|f_{\restriction \Delta} - f'_{\restriction \Delta}\|_{L^\infty(\Delta)}.$$

Based on the result, we can bound $\log \mathcal{N}(\varepsilon, \mathcal{F}_{DNN}^{S_n}, \|\cdot\|_{L^\infty(I)})$ by $\log \mathcal{N}(\varepsilon, \mathcal{F}_\Delta, \|\cdot\|_{L^\infty(\Delta)})$. Suppose $\log \mathcal{N}(\varepsilon, \mathcal{F}_\Delta, \|\cdot\|_{L^\infty(\Delta)}) =: K$ is finite. Then, there exist $f_{\restriction \Delta 1}, ..., f_{\restriction \Delta K}$, and for any $f_{\restriction \Delta} \in \mathcal{F}_\Delta$, there exists $j \in \{1, ..., K\}$ such as $\|f_{\restriction \Delta} - f_{\restriction \Delta j}\|_{L^\infty(\Delta)} \leq \varepsilon$. Here, for any $f \in \mathcal{F}_{DNN}^{S_n}$, there exists $f_j := f_{\restriction \Delta j} \circ g \in \mathcal{F}_{DNN}^{S_n}$ with corresponding $j$ and it satisfies $\|f - f_j\|_{L^\infty(I)} \leq \|f_{\restriction \Delta} - f_{\restriction \Delta j}\|_{L^\infty(\Delta)} \leq \varepsilon$. Then, we obtain the statement. $\qquad\square$

The second step of this section is shown by the following proposition:

**Proposition 5.** *With an existing constant $c > 0$ and $C$ in Lemma 1, for any $\delta > 0$, we obtain*

$$\log \mathcal{N}(2C_\Delta \delta, \mathcal{F}_{DNN}^\Delta, \|\cdot\|_{L^\infty(I)}) \leq \frac{C}{n! \, \delta^p} + \log\left(\frac{2cB}{\delta}\right).$$

Importantly, the result shows that the main term of the covering number is improved by $n!$, and it is a key factor to improve the overall generalization error.

*Proof of Poposition 5.* We bound a covering number of a set of $C_\Delta$-Lipschitz continuous functions on $\Delta$. Let $\{x_1, ..., x_K\} \subset \Delta$ by a set of centers of $\delta$-covering set for $\Delta$. By Lemma 1, we set $K = C/(n! \, \delta^p)$ with $\delta$ with a parameter $\delta > 0$, where $C > 0$ is a constant.

We will define a set of vectors to bound the covering number. We define a discretization operator $A : \mathcal{F}_\Delta \to \mathbb{R}^K$ as

$$Af = (f(x_1)/\delta, ..., f(x_K)/\delta)^\top.$$

Let $\mathcal{B}_\delta(x)$ be a ball with radius $\delta$ in terms of the $\|\cdot\|_\infty$-norm. For two functions $f, f' \in \mathcal{F}_\Delta$ such as $Af = Af'$, we obtain

$$\|f - f'\|_{L^\infty(I)} = \max_{k=1,...,K} \sup_{x \in \mathcal{B}_\delta(x_k)} |f(x) - f'(x)|$$

$$\leq \max_{k=1,...,K} \sup_{x \in \mathcal{B}_\delta(x_k)} |f(x) - f(x_k)| + |f'(x_k) - f(x_k)| \leq 2C_\Delta \delta,$$

where the second inequality follows $f(x_k) = f'(x_k)$ for all $k = 1, ..., K$ and the last inequality follows the $C_\Delta$-Lipschitz continuity of $f$ and $f'$. By the relation, we can claim that $\mathcal{F}_\Delta$ is covered by $2C_\Delta \delta$ balls whose center is characterized by a vector $b \in \mathbb{R}^K$ such as $b = Af$ for $f \in \mathcal{F}_\Delta$. Namely, $\mathcal{N}(2C_\Delta \delta, \mathcal{F}_\Delta, \|\cdot\|_{L^\infty(I)})$ is bounded by a number of possible $b$.

Then, we construct an explicit set of $b$ to cover $\mathcal{F}_\Delta$. Without loss of generality, assume that $x_1, ..., x_K$ are ordered satisfies such as $\|x_k - x_{k+1}\|_\infty \leq 2\delta$ for $k = 1, ..., K - 1$. By the definition, $f \in \mathcal{F}_\Delta$ satisfies $\|f\|_{L^\infty(\Delta)} \leq B$. $b_1 = f(x_1)$ can take values in $[-B/\delta, B/\delta]$. For $b_2 = f(x_2)$, since $\|x_1 - x_2\|_\infty \leq 2\delta$ and hence $|f(x_1) - f(x_2)| \leq 2C_\Delta \delta$, a possible value for $b_2$ is included in $[(b_1 - 2\delta)/\delta, (b_1 + 2\delta)/\delta]$. Hence, $b_2$ can take a value from an interval with length 4 given $b_1$. Recursively, given $b_k$ for $k = 1, ..., K - 1$, $b_{k+1}$ can take a value in an interval with length 4.

Then, we consider a combination of the possible $b$. Simply, we obtain the number of vectors is $(2cB/\delta) \cdot (4c)^{K-1}$ with a universal constant $c \geq 1$. Then, we obtain that

$$\log \mathcal{N}(2C_\Delta \delta, \mathcal{F}_\Delta, \|\cdot\|_{L^\infty}) \leq (K - 1) \log 4c + \log(2cB/\delta).$$

Then, we specify $K$ which describe a size of $\Delta$ through the set of covering centers. $\square$

*Proof of Theorem 2 and 3.* For Theorem 3, we combine the result in Proposition 4 and 5. For Theorem 2, we substitute the result in Theorem 3 into the well-known result (2), then obtain the statement. $\square$

### 4.3 PROOF FOR APPROXIMATION-CONTROL BOUND (THEOREM 4)

Proof of the approximation power also depends on the correspondence mapping $\Lambda$ in Proposition 2. Although Proposition 2 claims that the correspondence holds for a function by deep neural networks, the similar discussion in the proof shows that it holds for a general invariant function.

*Proof of Theorem 4.* Let $f^*$ be an invariant function on $I$. Then by Proposition 2, we have a function $f$ on $\Delta$ such that $f^* = f \circ \text{Sort}$ holds. By Theorem 5 in Schmidt-Hieber (2017), for enough big $N$, there exists a constant $c$ and a neural network $g$ with at most $\mathcal{O}(\log(N))$ layers and at most $\mathcal{O}(N \log(N))$ nonzero weights such that $\|f - g\|_{L^\infty(I)} \leq cN^{-\alpha/p}$. Then, we have

$$\|f^* - g \circ \text{Sort}\|_{L^\infty(I)} = \|f \circ \text{Sort} - g \circ \text{Sort}\|_{L^\infty(I)} = \|f - g\|_{L^\infty(\Delta)} \leq \|f - g\|_{L^\infty(I)} \leq cN^{-\alpha/p},$$

where $g \circ \text{Sort}$ is a neural network with at most $\mathcal{O}(\log(N)) + K_1$ layers and at most $\mathcal{O}(N \log(N)) + K_2$ nonzero weights, where $K_1$ and $K_2$ are the number of layers and the number of nonzero weights of the neural network expressing Sort respectively. By replacing $N^{-1}$ with $\varepsilon$, we have the desired inequality. $\qquad \square$

## 5 Discussion and Comparison

We discuss the technical non-triviality of our result. One can consider that our result seems straight-forward, because the improvement by $\sqrt{n!}$ looks a kind of folklore. However, there are several technical difficulties to prove it. Rigorously speaking, to obtain the improved bound, we have to find $n!$ subsets of functions *without* overlapping with each other. To find them, we introduce the notion of the fundamental domain (Definition 4) and prove that a volume of overlapping of the subsets has measure zero (Lemma 1). To the best of knowledge, this is the first study to show the result.

We mention that Sokolic et al. (2016) investigates a generalization bound for classification with invariant algorithms. Scope of the study is not limited to deep neural networks, but a wide class of learning problems. The generalization bound by the study is improved by $\sqrt{T}$, where $T$ is a number of possible *transformations* for invariance. To discriminate this paper from the study by Sokolic et al. (2016), we provide several differences between the study and this paper. Firstly, we construct an explicit framework for invariant deep neural networks, which guarantees the practical usage of several methods. Especially, the framework contains the famous network as the DeepSets by Zaheer et al. (2017). Hence, our paper can provide useful knowledge for practical use, while the study by Sokolic et al. (2016) investigates an abstract problem. Secondly, our analysis is not limited to classifications, but can be applied to general learning methods including regression. Thirdly, we focus on the more specific permutation invariance, and obtain the explicit improvement of the generalization bound. The bound by Sokolic et al. (2016) considers a more abstract problem with a wide class of invariance, hence it is not clear to obtain the same generalization bound of this paper.

## 6 Conclusion

In this paper, we develop a generalization theory to clarify the higher precision of the invariant deep neural network. Our generalization bound shows that it gets much tight by the invariant property, rigorously, the bound is improved by $\sqrt{n!}$ where $n$ is a number of permutation-invariant coordinates. Intuitively, this is caused by fact that the input space is divided into $n!$ copies of subspace which can be moved by permutation to each other. We further prove that the invariant deep neural network with a ReLU activation can achieve the optimal approximation rate for smooth functions. By the results, our theory shows a great advantage of deep neural networks.

As an improvement of our result, it is an open question to connect the invariant property and the norm-controlled entropy control for deep neural networks (e.g. the work by Bartlett et al. (2017)). To describe the practical high accuracy of deep learning, numerous studies investigigate the the norm-controlled entropy. We guess that our theory is valid with the entropy and more suitable to analyze the performance of invariant deep neural networks.

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
