# OpenReview forum: "Improved Generalization Bound of Permutation Invariant Deep Neural Networks"
_ICLR.cc/2020/Conference — Reject_

### Official Review · AnonReviewer1 · 2019-10-24
**Official Blind Review #1**

**Rating:** 6

**Review:**

This paper derives a generalization bound for permutation invariant networks. The main idea is to prove that the bound is inversely proportional to the square-root of the number of possible permutations to the input. The key result is Theorem 3 that bounds the covering number of a neural network (defined under an approximation control bound, Thm 4) using the number of permutations. The paper proves the theorem by showing that the space of input permutations can reduced to group actions over a fundamental domain, and deriving a bound for the covering number of the fundamental domain (Lemma 1), which is then extended to derive the same for the neural network setting. For the permutation invariance setting, the fundamental domain is obtained via the sorting operator.

Pros:
1. The paper appears to be mathematically rigorous, and at the same time, is straightforward to follow, with useful intuitions provided whenever required.
2. The provided theoretical result perhaps extends the work on universal approximation theorem for permutation invariant networks in Sennai et al, and Maron et al., 2019. Further, the generalization bound for permutation invariance is new to my knowledge.

Cons:
1. While, the proof appears to be novel for permutation invariance per se, however I do not think the main findings in this paper or the proof approach are sufficiently novel. For example, generalization bounds under invariances have been explored previously, perhaps the most related to this paper is [a] below that already shows (in a similar vein as this paper) that the bound decreases proportional to 1/\sqrt(T), where T is the number of invariances used. While, that work uses affine transformations of the input from a base space for the invariances (which this paper calls fundamental domain), the current paper uses permutation invariance and thus gets the bound proportional to 1/sqrt(n!). In the context of this prior work, the contribution of this paper appears incremental. The paper should cite this work and contrast against the results and proof methods in it.

[a] Generalization Error of Invariant Classifiers, Sokolic et al., ICML 2017.

2. The paper has several typos and grammatical errors through out, which are easily fixable though!

Overall, this paper is technically rigorous, and novel in its very specific context of deriving the generalization bounds for permutation invariant networks. However, in the broader context of invariances in general and their bounds, the contribution appears to be marginal.

**Experience Assessment:**

I do not know much about this area.

**Review Assessment: Checking Correctness Of Derivations And Theory:**

I assessed the sensibility of the derivations and theory.

**Review Assessment: Checking Correctness Of Experiments:**

N/A

**Review Assessment: Thoroughness In Paper Reading:**

I read the paper at least twice and used my best judgement in assessing the paper.

---

> ### Author Response · Authors · 2019-11-12
> **Thank you for your accurate comment.**
>
> Thank you for your accurate comment. Especially, we would appreciate your evaluation for our technical contributions.
>
> We also thank you for the introduction of the previous research[a].  We confirmed that their main result is very similar to ours. The superiority of our results is as follows. At first, we construct explicit invariant deep neural networks, which guarantee practical and useful methods. One of them is a new one that can achieve the same objectives as DeepSets (Zaheer 2018). Since the paper [a] is written with an abstract framework, our paper can provide useful knowledge. Secondly, our analysis is not limited to classification but can be applied to general learning methods including regression. Thirdly, our results provide a more specific analysis of permutation invariant networks, which can be used for future specific expansion and analysis.

---

> ### Author Response · Authors · 2019-11-15
> **We add the comparison with the paper (Sokolic et al., ICML 2017) in our paper.**
>
> We clarify the difference between our paper and okolic et al., ICML 2017 in Section in the updated version of our paper. We are glad if you check the paragraph.

---

### Official Review · AnonReviewer3 · 2019-10-25
**Official Blind Review #3**

**Rating:** 1

**Review:**

This paper provides generalization bounds for permutation invariant neural networks where the learning problem is invariant to the permutation of input data.

Unfortunately, the technical value of the content and its novelty is very limited since the proof reduces to a very basic argument that counts invariances (which is simply n! where n is the number of invariant dimensions) and uses a standard approach to give a generalization bound. Therefore, I don't think the results does not help us with better understanding of permutation invariant neural networks.

Unfortunately, the paper has several typos and mistakes as well. Another non-technical issue is that apparently authors have removed the ICLR format and reduced margin to fit the paper in 10 pages which is against the spirit of page limit.

***********************************

After author rebuttals:

After reading authors' response and reading the proofs, I realize that the formal proof is not trivial and requires more work that I assumed. However, I do not understand how this work can improve our understanding of permutation invariant networks. Therefore, I think the contributions are not significant enough for publication and my evaluation remains the same.

**Experience Assessment:**

I have published in this field for several years.

**Review Assessment: Checking Correctness Of Derivations And Theory:**

I assessed the sensibility of the derivations and theory.

**Review Assessment: Checking Correctness Of Experiments:**

N/A

**Review Assessment: Thoroughness In Paper Reading:**

I read the paper at least twice and used my best judgement in assessing the paper.

---

> ### Author Response · Authors · 2019-11-12
> **Could you give me some evidence or references?**
>
> Thank you for your comment.
>
> As mentioned in our paper, we developed several novel techniques as follow: (i) we prove a correspondence between invariant DNNs and DNNs on the fundamental domain, and (ii) we derive a covering number for a functional space which is sensitive to a volume of the domain of functions. To the best of our knowledge, such techniques are not commonly used in the analysis of deep neural networks.
>
> Could you give me some pieces of evidence or references which support your opinion that says our analysis follows very basis arguments? As your comment does not provide such clear evidence, we cannot find a way to discuss it with you.
>
> About the format of our paper, we mistakenly load the "fullpage" package, hence the margin of our paper is changed. About the point, we have not refutation and will modify it.

---

> > ### Comment · AnonReviewer3 · 2019-11-13
> > **Thanks for your reponse**
> >
> > I agree with you that generalization of invariant DNNs are not studied before. However, my main concern is the significance of the work. Basically, covering numbers count the number of different functions in the hypothesis class where the notion of different depends on some metric. Now, if there the input has invariance, one can take advantage of that and reduce this total number of different functions by n! Even though this very specific problem has not been studied before, it is not clear to me that this contribution is significant enough to be accepted at ICLR.

---

> > > ### Author Response · Authors · 2019-11-14
> > > **Thank you for your response.**
> > >
> > > Thank you for agreeing on the novelty of our work.
> > >
> > > About significance, we are confident that it is not easy to develop proof to derive the bound improved by n!. Technically speaking, to obtain the improved bound, we have to find n! subsets of functions WITHOUT overlapping with each other. To the aim, we introduce the notion of the fundamental domain and prove that a volume of overlapping has measure zero (Specifically, Lemma 1 in our paper). Without our techniques, the improvement by n! is a folklore, but not theoretical analysis. Hence, we believe that it is significant to develop such the technique and show the improved bound.
> > > If you are not agree with the importance of our achievement, please give us references which show the improvement rigorously.

---

### Official Review · AnonReviewer2 · 2019-10-29
**Official Blind Review #2**

**Rating:** 3

**Review:**

This paper presents a derivation of a generalization bound for neural networks designed specifically to deal with permutation invariant data (such as point clouds). The heart of the contribution is that the bound includes a  1/n! (i.e. 1 / (n-factorial)) factor to the major term, where n is the number of permutable elements there are in a data example (think: number of points in a point cloud). This term goes some way towards making the bound tight.

The 1/n! factor in the bound may be an interesting development  but the novelty does appear to be limited.  Also, the authors fail to discuss that --  as part of that same term -- there is a factor: (1 / (epsilon^p)), where p is the dimension of the input and epsilon is a small error term. As p is proportional to n, and epsilon is quite small, this term could well dominate the factorial in many practical settings. A discussion of the relation between these terms is appropriate and seems to be missing.

Clarity:
In general the paper is fairly well written, but there are multiple instances of missing articles and strange idiom violations (eg. p. 4, remark 1: "such the bound" versus "such a bound")

More seriously, the proof of Lemma 1 was quite hard to follow (esp. the second paragraph). I would suggest putting less emphasis on the relatively straightforward construction of the sorting mechanism in Propositions 2 and 3, and use the space to more clearly detail the proof of Lemma 1, which is, after all, the heart of the contribution.

I also found the proof of proposition 4 too confusing to easily follow. What is the interpretation of the indices (1, ..., K) on the functions?

Finally, I would have liked to see some interpretation of the findings in a discussion section (or in an extended conclusion).

Minor issues:

- First sentence of the abstract is difficult to parse and does not seem like an accurate assessment of the contribution of the paper.

- Paragraph 2 of the introduction presents a sequence of argument whose logic seems inconsistent to me. There is a drift from a discussion of generalization of neural networks to a mention of work on the very distinct topic of the representational capacity of neural networks (i.e. universal approximation property of neural networks). The linking text "To tackle the quesiton, ..." is not appropriate.

- Unlike Example 1, Example 2 (p.3) is not helpful in motivating the permutation invariant neural networks. The definition makes direct reference to Proposition 2 that will not be introduced for another 3 pages.

- In Sec. 4.1, it seems like a phi symbol is used when I believe a null symbol was intended

- Proposition 3: "max( z_1, z_1 )"  should be "max( z_1, z_2 )" with the adjustment carrying through to the other side of the equals.


**Experience Assessment:**

I do not know much about this area.

**Review Assessment: Checking Correctness Of Derivations And Theory:**

I assessed the sensibility of the derivations and theory.

**Review Assessment: Checking Correctness Of Experiments:**

N/A

**Review Assessment: Thoroughness In Paper Reading:**

I read the paper thoroughly.

---

> ### Author Response · Authors · 2019-11-12
> **Though the rate of $\varepsilon$ is critical, the generalization bound can be tight with large $n$.**
>
> Thank you for your critical opinion.
>
> As you mentioned, the order of $\varepsilon$ is very important, thus we will add the discussion. We expect that our generalization bound may get loose when $n$ is not sufficiently large. In contrast, when $n$ is reasonably large, our bound becomes tight since $n!$ increases rapidly rather than $\varepsilon^p$.
>
> About the clarity, we will modify our description and correct the typos.

---

> ### Author Response · Authors · 2019-11-15
> **We add discussion about $\varepsilon^p$.**
>
> We appreciate your critical comment.
> In our updated paper, we add description about the point after Theorem 2 in our paper. For summary, the increasing speed of $n! \varepsilon^p=n! \varepsilon^{nD}$ in terms of $n$ is sufficiently fast for any $\varepsilon$ and $D$. We appreciate if you check the point.

---

### Author Response · Authors · 2019-11-15
**We updated the submitted paper.**

Updated points are as follow:
- Add description to show the technical novelty and intuition of our paper. (Section 5 and 6)
- We cite the paper Sokolic+ (2017) and discuss differences between it and our paper. (Section 5)
- We show that the term $\varepsilon^p$ does not provide a problem with a large $n$. (Section 3)
- Correct several sentences and typos.
- We omit the mistakenly loaded package and modify the format as following the template.

---

### Decision · Program_Chairs · 2019-12-19

**Decision:**

Reject

**Comment:**

This work proves a generalization bound for permutation invariant neural networks (with ReLU activations). While it appears the proof is technically sound and the exact result is novel, reviewers did not feel that the proof significantly improves our understanding of model generalization relative to prior work. Because of this, the work is too incremental in its current form.